# Online Market Equilibrium with Application to Fair Division

**Yuan Gao**
Columbia University
gao.yuan@columbia.edu

**Alex Peysakhovich**
Facebook AI Research
alex.peys@gmail.com

**Christian Kroer**
Columbia University
christian.kroer@columbia.edu

## Abstract

Computing market equilibria is a problem of both theoretical and applied interest. Much research to date focuses on the case of static Fisher markets with full information on buyers' utility functions and item supplies. Motivated by real-world markets, we consider an online setting: individuals have linear, additive utility functions; items arrive sequentially and must be allocated and priced irrevocably. We define the notion of an online market equilibrium in such a market as time-indexed allocations and prices which guarantee buyer optimality and market clearance in hindsight. We propose a simple, scalable and interpretable allocation and pricing dynamics termed as PACE. When items are drawn i.i.d. from an unknown distribution (with a possibly continuous support), we show that PACE leads to an online market equilibrium asymptotically. In particular, PACE ensures that buyers' time-averaged utilities converge to the equilibrium utilities w.r.t. a static market with item supplies being the unknown distribution and that buyers' time-averaged expenditures converge to their per-period budget. Hence, many desirable properties of market equilibrium-based fair division such as envy-freeness, Pareto optimality, and the proportional-share guarantee are also attained asymptotically in the online setting. Next, we extend the dynamics to handle quasilinear buyer utilities, which gives the first online algorithm for computing first-price pacing equilibria. Finally, numerical experiments on real and synthetic datasets show that the dynamics converges quickly under various metrics.

## 1 Introduction

A market is said to be in equilibrium when supply is equal to demand. Computing prices and allocations which constitute a market equilibrium (ME) has long been a topic of interest [17, 20, 28, 31, 38, 43]. Most existing work focuses on the case of static markets. However, in this paper we consider the case of online markets where items arrive sequentially. We consider the extension of market equilibrium to this setting and provide market dynamics which quickly converge to an equilibrium in the case of online Fisher markets.

In static Fisher markets there is a fixed supply of each item, individual preferences are linear, additive, and items are divisible (or equivalently, randomization is allowed so individuals can purchase not just items but lotteries over items). In general, finding market equilibria is a hard problem [14, 39, 47]. However, in static linear Fisher markets, equilibrium prices and allocations can be computed via solving the Eisenberg-Gale (EG) convex program [22, 37].

We consider an online extension of Fisher markets where buyers are constantly present but items arrive one-at-a-time. Buyers' budgets are per-period and represent their respective 'bidding powers' instead of being binding constraints. We extend the definition of market equilibrium to the online setting: online equilibrium allocations and prices are time-indexed and, when averaged across time, form an equilibrium in a corresponding static Fisher market where item supplies are proportional

35th Conference on Neural Information Processing Systems (NeurIPS 2021).

to item arrival probabilities. Due to the stochastic nature of online Fisher markets, any online algorithm can only attain an online market equilibrium *asymptotically*, that is, the allocations and prices approximately satisfy the equilibrium conditions after running the algorithm for a long time.

We propose market dynamics that find these equilibria in an online fashion based on the *dual averaging* algorithm applied to a reformulation of the dual of the EG convex program. We refer to this mechanism as **PACE** (Pace According to Current Estimated utility). In PACE, each buyer is assigned a utility *pacing* multiplier at time $0$. When an item arrives, the individual with the highest adjusted utility (its valuation times the multiplier) receives that item and pays a price equal to its adjusted utility. The pacing multipliers of all individuals are then adjusted according to a closed-form rule which is given by the time average of the subgradient of the dual of the EG program. Intuitively, the pacing multipliers of those that did not receive the item go up while the receiver's typically (but not always) goes down. We show that PACE yields item allocations and prices that satisfy various equilibrium properties asymptotically, for example no-regret and envy-freeness.

One important application of market equilibrium is fair allocation using the *competitive equilibrium from equal incomes* (CEEI) mechanism [12, 46]. In CEEI, each individual is given an endowment of faux currency and reports her valuations for items; then, a market equilibrium is computed and the items are allocated accordingly. However, many fair division problems are online rather than static. These include the allocation of impressions to content in certain recommender systems [34], workers to shifts, donations to food banks [2], scarce compute time to requestors [25, 29, 40], or blood donations to blood banks [32]. Similarly, online advertising can also be thought of as the allocation of impressions to advertisers via a market though with a budget of real money rather than faux currency. In the static CEEI case with linear additive preferences, the resulting equilibrium outcomes (i.e. results of the EG program) have been described as "perfect justice" [3]. In the online case, PACE achieves the same fair allocations as CEEI asymptotically. See Appendix A for more related work in the areas of (static and online) equilibrium computation and fair division.

We evaluate PACE experimentally in several market datasets. Convergence to good outcomes happens quickly in experiments. Taken together our results, we conclude that PACE is an attractive algorithm for both computing online market equilibria and online fair division.

**Main contributions.** We consider the problem of allocating and pricing sequentially arriving items to $n$ buyers. This setting is termed as an *online Fisher market*. Given a sequence of item arrivals, we define an online market equilibrium as the items' allocations and prices that, in hindsight, ensure buyer optimality and market clearance. We propose the PACE dynamics, which can be viewed as a nontrivial instantiation of the dual averaging algorithm on a reformulation of the dual of the Eisenberg-Gale convex program. Leveraging the convergence theory of dual averaging, we show that, when item arrivals are drawn from an (unknown) underlying distribution $s$, possibly over an infinite/continuous item space, PACE ensures the following.

- The pacing multipliers generated by PACE converge to the static equilibrium *utility prices*. Here, "static" means w.r.t. to an underlying static Fisher market.
- Buyers' time-averaged utilities converge to the static equilibrium utilities.
- Buyers' time-averaged expenditures converge to their respective budgets.

These convergences are all in mean square with rates $O((\log t)/t)$, $O((\log t)/t)$ and $O((\log t)^2/t)$, respectively, where the constants in these rates involve moderate polynomials of $n$. In this way, PACE generates allocations and prices that constitute an online market equilibrium in the limit. In particular, the allocations and prices ensure that the allocation is Pareto optimal, and buyers have no regret, no envy, and get at least their proportional share asymptotically. We also extend PACE to the case of quasilinear buyer utilities, which yields the first online algorithm for computing first-price pacing equilibria. Finally, numerical experiments suggest that PACE converges much faster than its theoretical rates in terms of pacing multipliers, utilities and expenditures.

## 2 Static and Online Fisher Markets

**Static Fisher markets and equilibria.** We first introduce static Fisher markets and their equilibria. Following the recent work [24, §2], we consider a measurable (possibly continuous) item space.

Below are the technical preliminaries for the subsequent online setting. They can be skimmed through and referred back to as needed.

From now on, we define $[k] := \{1, \ldots, k\}$ for any $k \in \mathbb{N} := \{0, 1, 2, \ldots\}$ and $\mathbb{R}_+$ ($\mathbb{R}_{++}$, resp.) as the set of nonnegative (positive, resp.) real numbers. Let $\mathbb{I}\{A\} \in \{0, 1\}$ denote the indicator function of an event $A$.

(a) There are $n$ buyers (individuals), each having a budget $B_i > 0$.

(b) The item space is a *finite* measure space $(\Theta, \mathcal{M}, \mu)$ with $0 < \mu(\Theta) < \infty$. From now on, $L^p$ (and $L^p_+$, resp.) denote the set of (nonnegative, resp.) $L^p$ functions on $\Theta$ for any $p \in [1, \infty]$ (including $p = \infty$). Below are some concrete special cases for illustration.

    (i) Finite: $\Theta = [m]$, $\mathcal{M} = 2^{[m]} = \{A : A \subseteq [m]\}$ and $\mu(A) = \sum_{a \in A} \mu(a)$ (all $2^m$ subsets are measurable and the measure is given by a point mass on each item).

    (ii) Lebesgue-measurable: $\mu$ is the Lebesgue measure on $\mathbb{R}^d$, $\mathcal{M}$ is the Lebesgue $\sigma$-algebra and $\Theta$ is a (Lebesgue-)measurable subset of $\mathbb{R}^d$ with positive finite measure. For example, $\Theta$ can be a compact subset of $\mathbb{R}^d$ with a nonempty interior.

    (iii) Countably infinite: $\Theta = \mathbb{N}$ and $\mu(A) = \sum_{a \in A} \mu(a)$ for any $A \subseteq \mathbb{N}$, where $\mu(\mathbb{N}) < 0$. For example, $\mu(a)$ can be the probability mass of a Poisson distribution, in which case $(\mathbb{N}, \mathcal{M}, \mu)$ is a probability space.

(c) The *supplies* of items is $s \in L^\infty_+$, i.e., item $\theta \in \Theta$ has supply $s(\theta)$. Since $\Theta$ is compact, it is measurable with a finite measure. For the finite case $\Theta = [m]$, we have $s = (s_1, \ldots, s_m) \in \mathbb{R}^m_+$.

(d) The *valuation* of each buyer $i$ on all items is $v_i \in L^1_+$, i.e., buyer $i$ has valuation $v_i(\theta)$ on item $\theta \in \Theta$. For the finite case $\Theta = [m]$, we have $v_i = (v_{i1}, \ldots, v_{im}) \in \mathbb{R}^m_+$.

(e) For buyer $i$, an *allocation* of items $x_i \in L^\infty_+$ gives a utility of

$$u_i(x_i) := \langle v_i, x_i \rangle := \int_\Theta v_i(\theta) x_i(\theta) d\theta,$$

where the angle brackets are based on the notation of applying a bounded linear functional $x_i$ to a vector $v_i$ in the Banach space $L^1$ and the integral is the usual Lebesgue integral. For the finite case $\Theta = [m]$, we have $x_i = (x_{i1}, \ldots, x_{im}) \in \mathbb{R}^m_+$ and the utility is

$$u_i(x_i) = \langle v_i, x_i \rangle = \sum_j v_{ij} x_{ij},$$

the usual Euclidean vector inner product. We will use $x \in (L^\infty_+)^n$ to denote the aggregate allocation of items to all buyers, i.e., the concatenation of all buyers' allocations.

(f) The *prices* of items are modeled as $p \in L^1_+$; in other words, the price of item $\theta \in \Theta$ is $p(\theta)$. For the finite case $\Theta = [m]$, we have $p = (p_1, \ldots, p_m) \in \mathbb{R}^m_+$.

(g) For a measurable item subset $A \subseteq \Theta$, let $v_i(A) := \int_A v_i(\theta) d\theta$ (and similarly for $p$ and $s$), the $v_i$-induced measure of $A$. For the finite case $\Theta = [m]$, for any item subset $A \subset [m]$, $v_i(A) = \sum_{j \in A} v_{ij}$ (and similarly for $p(A)$ and $s(A)$).

(h) Without loss of generality, we assume a unit total budget $\|B\|_1 = 1$, a unit total supply $s(\Theta) = 1$ and normalized buyer valuations $\langle v_i, s \rangle = 1$. In other words, all items have a total value of 1 for every buyer.

**Definition 1.** *Given item prices $p \in L^1_+$, the **demand** of buyer $i$ is its set of utility-maximizing allocations given the prices and budget:*

$$D_i(p) := \arg\max\{\langle v_i, x_i \rangle : x_i \in L^\infty_+, \langle p, x_i \rangle \leq B_i\}.$$

*The associated **utility level** $\hat{U}_i(p)$ is defined as the value of $\langle v_i, x_i \rangle$ for any $x_i \in D_i(p)$.*

**Definition 2.** *A **market equilibrium (ME)** is an allocation-price pair $(x^*, p^*) \in (L^\infty_+)^n \times L^1_+$ such that the following holds.*

    *(i) Supply feasibility: $\sum_i x_i^* \leq s$.*

    *(ii) Buyer optimality: $x_i^* \in D_i(p^*)$ for all $i$.*

*(iii) Market clearance:* $\langle p^*, s - \sum_i x_i^* \rangle = 0$ *(any item with a positive price is fully allocated).*

In the above definition and subsequently, all equations involving measurable functions are understood as "holding almost everywhere." For example, $\sum_i x_i \leq s$ means the (measurable) set $\{\theta \in \Theta : \sum_i x_i(\theta) \leq s(\theta)\}$ has the same measure as $\Theta$. Given a ME $(x^*, p^*)$, we often denote the (unique) equilibrium utilities as $u_i^* = \langle v_i, x_i^* \rangle$. For a finite-dimensional linear Fisher market, it is well known that a ME can be computed via solving the EG convex program. Recently, [24] generalized this framework to handle the case of an infinite item space. More specifically, consider the following (possibly infinite-dimensional) convex programs.

$$\sup_{x \in (L_+^\infty)^n} \sum_i B_i \log\langle v_i, x_i \rangle \text{ s.t. } \sum_i x_i \leq s. \qquad (\mathcal{P}_{\text{EG}})$$

$$\inf_{p \in L_+^1, \beta \in \mathbb{R}_+^n} \left( \langle p, s \rangle - \sum_i B_i \log \beta_i \right) \text{ s.t. } p \geq \beta_i v_i, \ \forall\, i. \qquad (\mathcal{D}_{\text{EG}})$$

The following theorem summarizes the results in [24, §3] regarding the above convex programs capturing market equilibria. As shown in that work, the above convex programs satisfy strong duality and their optimal solutions (which correspond to ME) can be characterized by the KKT optimality conditions. We slightly generalize the assumptions of [24] by allowing non-uniform item supplies $s$ instead of $s(\theta) = 1$ for all $\theta \in \Theta$. For completeness, a proof, which is mainly based on the proofs of the results in [24, §3], can be found in the Appendix.

**Theorem 1.** *The following hold regarding* $(\mathcal{P}_{\text{EG}})$ *and* $(\mathcal{D}_{\text{EG}})$.

- *Both suprema are attained.*
- *Given $x^*$ feasible to $(\mathcal{P}_{\text{EG}})$ and $(p^*, \beta^*)$ feasible to $(\mathcal{D}_{\text{EG}})$, they are both optimal if and only if the following holds: (i) $\langle p^*, s - \sum_i x_i^* \rangle = 0$ (market clearance), (ii) $\langle p^* - \beta_i^* v_i, x_i^* \rangle = 0$ (buyer $i$ only receives items within its 'winning set' $\{p^* = \beta_i^* v_i\}$) (ii) and $\langle v_i, x_i^* \rangle = u_i^* := B_i / \beta_i^*$ (buyer $i$ gets its maximum possible utility from $x_i^*$). In this case, $(x^*, p^*)$ is a ME.*
- *Conversely, for a ME $(x^*, p^*)$, it holds that (i) $x^*$ is an optimal solution of $(\mathcal{P}_{\text{EG}})$ and (ii) $(p^*, \beta^*)$, where $\beta_i^* := B_i / \langle v_i, x_i^* \rangle$, is an optimal solution of $(\mathcal{D}_{\text{EG}})$.*

In the above theorem, $\beta_i^*$ is known as buyer $i$'s *utility price*, i.e., price per unit utility at equilibrium. As is well known, in a ME $(x^*, p^*)$, the allocations $x^*$ are

inline Pareto optimal,

inline envy-free (in a budget weighted sense, i.e., $\langle v_i, x_i^* \rangle / B_i \geq \langle v_i, x_k^* \rangle / B_k$ for all $k \neq i$),

inline proportional (i.e., $\langle v_i, x_i^* \rangle \geq \langle v_i, s \rangle / n = 1/n$); see, e.g.,[24, Theorem 3].

**Online Fisher markets and equilibria.** We now consider a simple online variant of the Fisher market setting, referred to as an online Fisher market (OFM). There are $n$ buyers, each with a valuation $v_i \in L_+^1$. Assume there are discrete time steps $t = 1, 2, \ldots$. At each time step $t$, an item $\theta_t$ arrives and each buyer $i$ sees a value $v_i(\theta_t)$. The item must be allocated irrevocably to one buyer. Each buyer $i$ has a budget $B_i > 0$ representing her per-period expenditure rate. [1]

Next, we introduce the notions of demand, utility level and online market equilibrium in an OFM. All of them are defined based on sequences of arrived items and their prices; they do not require any distributional assumption on the item arrivals.

**Definition 3.** *Let the arrived items be $(\theta_\tau)_{\tau \in [t]}$. An allocation (of arrived items) is $(x_i^\tau)_{(\tau,i) \in [t] \times [n]}$, where $x_i^\tau \in [0,1]$ is the fraction of the item $\theta_\tau$ allocated to buyer $i$.[2] Let the prices of the arrived*

---

[1]This assumption is similar to one made in the literature on budget management in auctions, where each buyers has a per-period expenditure rate and the overall budget equal to the rate times the number of time periods. If a hard budget cap across all time periods is desired, then PACE and similar mechanisms may deplete some buyers' budgets close to the end of the horizon [6–8].

[2]We allow fractional allocations in the definition for more generality. As we will see, fractional allocation is not needed: PACE generates allocations and prices that satisfy the OME conditions asymptotically via assigning each arrived item to one buyer.

*items be $(p^\tau(\theta_\tau))_{\tau \in [t]}$. The* demand *of each buyer $i$ (in hindsight) at time $t$ is*

$$D_i^t = \arg\max_{(z_i^\tau)_{\tau \in [t]}} \left\{ \frac{1}{t} \sum_{\tau=1}^{t} v_i(\theta_\tau) z_i^\tau : 0 \leq z_i^\tau \leq 1, \forall \tau, \ \frac{1}{t} \sum_{\tau=1}^{t} p^\tau(\theta_\tau) z_i^\tau \leq B_i \right\}. \quad (1)$$

*Let $\hat{U}_i^t$ be the* utility level *associated with this demand, i.e., the maximum value in (1). An **online market equilibrium (OME)** is a pair of allocations $(x_i^\tau)_{(\tau,i) \in [t] \times [n]}$ and prices $p^\tau(\theta_\tau)$ such that the following holds.*

(i) *Total allocation does not exceed the unit amount of the item $\sum_i x_i^\tau \leq 1$ for all $\tau$.*

(ii) *Buyers' realized allocations are optimal in hindsight: $(x_i^\tau)_{\tau \in [t]} \in D_i^t$ for all $i$.*

(iii) *Market clearance: $\sum_i x_i^\tau = 1$ for $\tau$ such that $p^\tau(\theta_\tau) > 0$.*

In words, $\hat{U}_i^t$ is the maximum possible (time-averaged) utility buyer $i$ could have attained via choosing from the arrived items $(\theta_\tau)_{\tau \in [t]}$ in hindsight, subject to their respective posted prices $(p^\tau(\theta_\tau))_{\tau \in [t]}$ and her current total budget $tB_i$, with $D_i^t$ being the set of such utility-maximizing (time-indexed) allocations subject to per-period item availability constraints. An OME is a pair of allocations and prices that make buyers optimal in hindsight and market cleared.

Given an OFM, we define the associated underlying static Fisher market as having the same $n$ buyers and an item space $\Theta$ with supply $s$ being the (unknown) distribution from which the arriving items $\theta_t$ are drawn. To clarify the concepts of OFM and OME, we consider some simple special cases.

- Suppose all item arrivals $\theta_1, \ldots, \theta_t$ are known in advance. Then, the OFM is the same as a static $n \times t$ Fisher market with the same buyers and the $t$ items, each having a unit supply. Here, buyer $i$'s valuation of item $\tau$ is $v_{i\tau} = v_i(\theta_\tau)$. To compute an OME, it suffices to solve the classical (finite-dimensional) Eisenberg-Gale convex program, that is, $(\mathcal{P}_{\mathrm{EG}})$ with $\Theta = [t]$, $s = (1, \ldots, 1) \in \mathbb{R}_+^t$ and $x \in \mathbb{R}_+^{n \times t}$. Let the static ME be $(x^*, p^*) \in \mathbb{R}_+^{n \times t} \times \mathbb{R}_{++}^t$. When each item $\theta_\tau$ arrives, OME allocates a fraction $x_{i\tau}^*$ of the item to each buyer $i$ and set its price as $p_\tau^*$.

- Suppose the sequentially arriving items are drawn i.i.d. from a known underlying distribution $s \in L_+^\infty$ (which specifies a random variable $\theta \sim s$ such that $\mathbf{P}[\theta \in A] = s(A)$ for any measurable set $A \subseteq \Theta$) and all buyers' valuations $v_i$ are known. Suppose we have also computed a static ME $(x^*, p^*)$ (Definition 2) of a market with buyer valuations $v_i$, budgets $B_i$ and item supplies being the distribution $s$ (the *underlying static market*). Then, when a new item $\theta_t$ (which is drawn from the distribution $s$) arrives at time $t$, set its price as $p^*(\theta_t)$ and allocate a fraction $x_i^*(\theta_t)/s(\theta_t)$ of it to each buyer $i$ (assume $s(\theta_t) > 0$, i.e., only items with positive supplies can appear). Then, the time-averaged utility of each buyer $i$ is $\frac{1}{t} \sum_{\tau=1}^{t} v_i(\theta_t) x_i^*(\theta_t)/s(\theta_t)$, which converges to $\mathbf{E}_{\theta \sim s}[v_i(\theta) x_i(\theta)/s(\theta)] = \int_\Theta v_i(\theta) x_i^*(\theta) d\theta = u_i^*$ a.s. by to the Strong Law of Large Numbers. Since the online process is carried out using static equilibrium prices and allocations, the static ME properties (Definition 2) ensure the required OME properties hold asymptotically.

The above special cases require full knowledge of either the exact future item arrivals or the underlying static market to attain an OME. Next, we propose a simple, distributed dynamics which generates allocations and prices that satisfy the OME conditions asymptotically **without** requiring such knowledge (in particular, without knowledge of the distribution $s$).

## 3 The PACE Dynamics

In this section, we introduce the **PACE** (Pacing According to Current Estimated utility) dynamics that prices and allocates sequentially arriving items via (i) maintaining a *pacing multiplier* for each buyer and (ii) simple, distributed updates.[3] In §5, we will show that PACE is an instantiation of dual averaging [48], a stochastic first-order method for regularized optimization, applied to a reformulation of $(\mathcal{D}_{\mathrm{EG}})$. In the PACE dynamics, each buyer maintains a pacing multiplier $\beta_i^t$, starting from an initial value $\beta_i^1 = 1 + \delta_0$ for some small $\delta_0 > 0$ (e.g., $\delta_0 = 0.05$). At time step $t$, the following events take place.

---

[3]Pacing and pacing multipliers are terminology in budget management in large-scale ad auctions [18, 19].

(a) An item $\theta_t$ appears and each buyer $i$ sees a value $v_i(\theta_t)$ for the item.

(b) Each buyer $i$ bids their paced value $\beta_i^t v_i(\theta_t)$ for the item.

(c) The item is allocated to the highest bidder (the *winner* at $t$): $i_t = \arg\max_i \beta_i^t v_i(\theta_t)$, with ties broken arbitrarily. For concreteness, we always choose the lowest winning index, i.e.,

$$i_t = \min \arg\max_i \beta_i^t v_i(\theta_t).$$

Then, the price of $\theta_t$ is set by the first-price rule

$$p^t(\theta_t) = \max_i \beta_i^t v_i(\theta_t) = \beta_{i_t}^t v_i(\theta_t)$$

and the winner $i_t$ pays this price $p^t(\theta_t)$ for the item $\theta_t$.

(d) Each buyer $i$ gets a utility

$$u_i^t = v_i(\theta_t)\mathbb{I}\{i = i_t\}.$$

In other words, the winner $i_t$ gets $v_{i_t}(\theta_t)$ and other buyers get zero.

(e) Each buyer $i$ updates its cumulative average utility $\bar{u}_i^t$:

$$\bar{u}_i^t = \frac{1}{t}\sum_{\tau=1}^{t} u_i^\tau = \frac{t-1}{t}\bar{u}_i^{t-1} + \frac{1}{t}u_i^t.$$

(f) Each buyer $i$ updates their pacing multiplier $\beta_i^{t+1}$ as follows:

$$\beta_i^{t+1} = \Pi_{[l_i, h_i]}(B_i/\bar{u}_i^t) := \min\{\max\{l_i, B_i/\bar{u}_i^t\}, h_i\}.$$

where $l_i = B_i/(1+\delta_0)$ and $h_i = 1 + \delta_0$ for some fixed $\delta_0 > 0$ (e.g., $\delta_0 = 0.05$).

As will be seen in §5, buyer $i$'s equilibrium pacing multiplier (i.e., utility price) satisfies $l_i < \beta_i^* < h_i$ and her per-period utility $u_i^t$ corresponds to the $i$th component of a stochastic subgradient of a function on $\beta$ in a reformulation of the convex program ($\mathcal{D}_{\mathrm{EG}}$), on which we run dual averaging. Furthermore, the update rule for $\beta_i^{t+1}$ is such that, if the realized utilities $\bar{u}_i^t$ were the true static equilibrium utility for buyer $i$, then $\beta_i^{t+1}$ would be the equilibrium multiplier. Note that PACE does not randomize (any randomness can only come from the market environment from which item arrivals are drawn) and assigns every item to a single buyer without splitting it.

The simplicity and distributed nature of PACE makes it desirable for large-scale practical use.

- It can be run on arbitrary sequential item arrivals and only requires buyers' valuations $v_i(\theta_t)$ on the arrived items (rather than all valuations $v_i$ over the potentially large item space). No parameter tuning is needed (in particular, no stepsize tuning as in many first-order optimization methods).

- When run as a *centralized* allocation mechanism, PACE only needs to maintain $O(n)$ scalars, namely, $\beta_i^t$, $B_i$ and $\bar{u}_i^t$ for all $i$. At time $t$, it observes buyers' valuations $v_i(\theta_t)$ of the item $\theta_t$, compute bids $\beta_i^t v_i(\theta_t)$, finds the winner $i_t$, set the price as the maximal bid $\beta_{i_t}^t v_{i_t}(\theta_t)$ and allocates the item to the winner; finally, it updates $\bar{u}^t$ and $\beta^{t+1}$ as in (f), which takes $O(n)$ time.

- PACE can also be run among the buyers in a *decentralized* manner, in which case each buyer only maintains two scalar values: the pacing multiplier $\beta_i^t$ and time-averaged utility $\bar{u}_i^t$. When a new item arrives, each buyer only performs a few simple arithmetic operations to create a bid $\beta_i^t v_i(\theta_t)$, receives her utility (if she wins) and subsequently updates $\bar{u}_i^t$ and $\beta_i^{t+1}$.

These make PACE suitable for Internet-scale online fair division and online Fisher market applications. In particular, it is very reminiscent of how Internet advertising auctions are run. There, a similar auction-based system is used, with the pacing multiplier ensuring that each advertiser smooths out their budget expenditure across the many auctions. The primary difference between this and our setting is that (i) the auction can be first-price or second-price and (ii) buyers usually have *quasilinear* utilities, that is, utility of the item minus the expenditure (price paid) [6–8, 18]. In §C, we extend PACE to quasilinear utilities, which provides a novel online algorithm for first-price pacing equilibrium computation [19].

# 4 Dual Averaging

In this section, we briefly recap the setup and general convergence results of *dual averaging* [35, 48], which will be used in the analysis of PACE. First, we introduce some notation for this and the next section. Let $\mathbf{e}^{(i)}$ denote the $i$'th unit basis vector in $\mathbb{R}^n$ and $\mathbf{1} \in \mathbb{R}^n$ denote the vector of 1's. For $x, y \in \mathbb{R}^n$, $[x, y]$ denotes the Cartesian product of intervals $\prod_{i=1}^n [x_k, y_k] \subseteq \mathbb{R}^n$. All norms $\| \cdot \|$ without a subscript are Euclidean 2-norms, unless otherwise stated.

Let $\Psi$ be a closed, strongly convex function with domain $\operatorname{dom} \Psi := \{w \in \mathbb{R}^n : \Psi(w) < \infty\}$. Here, we do not employ any auxiliary regularizing function, since our problem has a natural source of strong convexity (i.e., a strongly convex $\Psi$) through the $-B_i \log \beta_i$ terms in ($\mathcal{D}_{\mathrm{EG}}$). Let $Z \subseteq \mathbb{R}^d$ be an arbitrary sample space. For each $z \in Z$, let $f_z$ be a convex and subdifferentiable function on $\operatorname{dom} \Psi$. Considers the following regularized convex optimization problem [48, §1.1]:

$$\min_w \mathbf{E} f_z(w) + \Psi(w), \tag{2}$$

where the expectation is taken over a probability distribution $\mathcal{D}$ on $Z$. We assume access to an oracle that, given any $f_t$ and $w \in \operatorname{dom} \Psi$, returns a subgradient $g^t \in \partial f_t(w)$. The dual averaging algorithm (DA) [48, Algorithm 1], with a strongly convex $\Psi$ and no auxiliary regularizer, is as follows. First, set $w_1 \in \arg\min_w \Psi(w)$ and $\bar{g}^0 = 0$. Then, for each $t = 1, 2, \ldots$, DA performs the following steps:

(1) Observe $f_t$ and compute $g^t \in \partial f_t(w^t)$.
(2) Update the average subgradient (the *dual average*) via $\bar{g}^t = \frac{t-1}{t} \bar{g}^{t-1} + \frac{1}{t} \bar{g}^t$.
(3) Compute the next iterate $w^{t+1} = \arg\min_w \{\langle \bar{g}^t, w \rangle + \Psi(w)\}$.

The following convergence guarantee on DA is proved as part of the proof of Corollary 4 in [48].

**Theorem 2.** *Dual averaging generates iterates $w^t$ such that*

$$\mathbf{E} \|w^t - w^*\|^2 \le \frac{(6 + \log t) G^2}{t \sigma^2},$$

*where $G^2$ is an upper bound on $\mathbf{E} \|g^t\|^2$, $t = 1, 2, \ldots$ and $\sigma$ is the strong convexity modulus of $\Psi$.*

When solving the stochastic optimization problem (2), in Theorem 2, we can set $G^2$ to be an upper bound on $\sup_{w \in \operatorname{dom} \Psi} \mathbf{E} \|g_z(w)\|^2$, where $g_z(w)$ is a subgradient oracle mapping each $(z, w) \in Z \times \operatorname{dom} \Psi$ to a subgradient and the expectation is over $z \sim \mathcal{D}$ and possible randomness of the subgradient oracle. We will shortly see that a reformulation of $\mathcal{D}_{\mathrm{EG}}$, when cast into the form (2), exhibits stochastic subgradients that are exactly buyers' received utilities in each time step. Using Theorem 2, we can show that the sequence of pacing multipliers $\beta^t$ generated by PACE converges to the underlying (equilibrium) utility prices $\beta^*$ of the static Fisher market.

# 5 Convergence Analysis of the PACE dynamics

We will now show that PACE correspond to running DA on the vector $\beta^t$ of pacing multipliers for the buyers. To this end, we first reformulate ($\mathcal{D}_{\mathrm{EG}}$) into a (finite-dimensional) convex program in $\beta$ in the form of (2):

$$\min_\beta \left( \left\langle \max_i \beta_i v_i, s \right\rangle - \sum_i B_i \log \beta_i \right) \quad \text{s.t. } \beta \in [B/(1+\delta_0), (1+\delta_0)\mathbf{1}], \tag{3}$$

where $\delta_0 > 0$ is an arbitrarily small constant. The bounds on $\beta$ do not change the optimal solution, because $\beta_i^* \in (B_i, 1)$ for each $i$. Detailed steps of the reformulation are given in Appendix B.

In order to run DA, we need to compute a subgradient of $f_\theta : \beta \mapsto \max_i \beta_i v_i(\theta)$ at any $\theta \in \Theta$. Following [24, §5], since $f_\theta$ is a piecewise linear function, a subgradient is

$$g_\theta(\beta) := v_{i^*}(\theta) \mathbf{e}^{(i^*)} \in \partial f_\theta(\beta),$$

where $i^* = \min \arg\max_i \beta_i v_i(\theta)$ is the winner (see, e.g., [10, Theorem 3.50]).

We can now show that the PACE dynamics corresponds to running DA on (3). Here, $\Psi(\beta) = -\sum_i B_i \log \beta_i$ with $\operatorname{dom} \Psi = [B/(1+\delta_0), (1+\delta_0)\mathbf{1}]$. First, choose $\beta^1 = \arg\min \Psi = (1+\delta_0)\mathbf{1}$ (i.e., $\beta_i^1 = 1 + \delta_0$ for all $i$) and $\bar{g}^0 = 0$. At each time step $t = 1, 2, \ldots$, given the current pacing multiplier $\beta^t$, DA applied to (3) unrolls the following steps.

- An item $\theta_t$ arrives, having value $v_i(\theta_t)$ for each buyer $i$. The function $f_t$ in DA is
$$f_{\theta_t} : \beta \mapsto \max_i \beta_i v_i(\theta_t).$$

- The winner is $i_t = \min \arg \max_i \beta_i^t v_i(\theta_t)$ and a subgradient is $g^t = v_{i_t j_t} \mathbf{e}^{(i_t)} \in \partial f_t(\beta^t)$. Its $i$th entry is exactly the realized (single-period) utility of individual $i$ at time $t$ in PACE, that is, $g_i^t = v_i(\theta_t) \mathbb{I}\{i = i_t\} = u_i^t$.

- Update the dual average (time-averaged utilities): for each $i$, compute $\bar{g}^t = \frac{t-1}{t} \bar{g}^{t-1} + \frac{1}{t} g^t$, i.e.,
$$\bar{g}_i^t = \frac{t-1}{t} \bar{g}_i^{t-1} + \frac{1}{t} v_i(\theta_t) \mathbb{I}\{i = i_t\}.$$

- Update the pacing multipliers:
$$\beta^{t+1} = \underset{\beta \in [B/(1+\delta_0),(1+\delta_0)\mathbf{1}]}{\arg\min} \left\{ \langle \bar{g}^t, \beta \rangle - \sum_i B_i \log \beta_i \right\}.$$

The minimization problem is separable in each $i$ and exhibits a simple and explicit solution which recovers step (f) in PACE (where $\bar{g}_i^t = \bar{u}_i^t$):
$$\beta_i^{t+1} = \underset{\beta_i \in [B/(1+\delta_0),1+\delta_0]}{\arg\min} \left\{ \bar{g}_i^t \beta_i - B_i \log \beta_i \right\} \Rightarrow \beta_i^{t+1} = \Pi_{[B_i/(1+\delta_0),1+\delta_0]} \left( \frac{B_i}{\bar{u}_i^t} \right).$$

As mentioned earlier, PACE does not require a stepsize parameter. This is because DA is stepsize-free given a strongly convex regularizer $\Psi$, which is indeed the case in our reformulation (3). In addition, in the above update step for $\beta_i^{t+1}$, the directions of change are as follows.

- For a non-winner $i \neq i_t$, we have $u_i^t = 0$ and hence $\bar{u}_i^t \leq \bar{u}_i^{t-1}$. This implies $\beta_i^{t+1} \geq \beta_i^t$. In words, a non-winner's pacing multiplier weakly increases. The increase is strict if $\bar{u}_i^{t-1} > 0$, i.e., buyer $i$ has already received a nonzero utility.

- For the winner $i_t$, $\bar{u}_{i_t}^t$ may become greater than $\bar{g}_{i_t}^{t-1}$, in which case $\beta_{i_t}^{t+1} \leq \beta_{i_t}^t$. In words, the winner's pacing multiplier may go up or down.

In order to analyze PACE, we assume $v_i(\Theta) = 1$, $v_i \in L_+^\infty$ (normalized and a.e.-bounded valuations)[4] and that there is an underlying item distribution $s \in L_+^\infty$ from which the item arrivals $\theta_t$, $t = 1, 2, \ldots$ are drawn i.i.d.[5] Define the underlying static Fisher market as one having the same $n$ buyers (each with valuation $v_i$ and budget $B_i$) and item supplies $s$. Denote the equilibrium utilities and utility prices w.r.t. the underlying static market as $u^*$ and $\beta^*$, respectively. We further assume that the valuations are $v_i \in L_+^\infty$ (i.e., a.e.-bounded on the item space). This is not restrictive: since an individual item $\theta$ has value $v_i(\theta)$ for each buyer $i$, it should be a finite value.

**Convergence of pacing multipliers.** After aligning PACE with DA, the convergence of the pacing multipliers $\beta^t$ follows directly from Theorem 2.

**Theorem 3.** *PACE generates pacing multipliers $\beta^t$ such that*
$$\mathbf{E}\|\beta^t - \beta^*\|^2 \leq \frac{(6 + \log t)G^2}{t\sigma^2}, \quad t = 1, 2, \ldots,$$
*where $G^2 = \max_i \mathbf{E}_{\theta \sim s}[v_i(\theta)^2] \leq \max_i \|v_i\|_\infty^2$, $\sigma = \frac{\min_i B_i}{(1+\delta_0)^2}$.*

In other words, we have mean-square convergence of $\beta^t$ to $\beta^*$ at a $O((\log t)/t)$ rate. Since $\|B\|_1 = 1$, we have $\min_i B_i \leq 1/n$. Hence, $\sigma = O(1/n)$ and the constant in the bound is $\Omega(n^2)$. Whether such dependence on $n$ can be improved via new analysis remains an interesting research question.

**Convergence of utilities.** We next show that the time-averaged utility $\bar{u}^t$ (which equals to the dual average $\bar{g}^t$) converges to the equilibrium utility vector $u^*$ of the underlying Fisher market. A key step in the proof is to bound the probability of a projection in updating $\beta_i^{t+1}$, that is, $\mathbf{P}[B_i/\bar{u}_i^t \notin [l_i, u_i]]$.

---

[4]The a.e.-boundedness assumption is needed in subsequent convergence analysis. Since $\Theta$ has a finite measure, it holds that $L_+^\infty \subseteq L_+^1$. For a finite item space $\Theta = [m]$, both are equal to $\mathbb{R}_+^m$.

[5]The distributional assumption on item arrivals (i.e., they are drawn i.i.d. from an unknown distribution $s$) is needed to establish asymptotic equilibrium properties of PACE. See Appendix B for an example that any algorithm can yield arbitrarily suboptimal allocations without such a distributional assumption.

**Theorem 4.** *For each $i$, let $\epsilon_i := \min\{h_i - \beta_i^*, \beta_i^* - l_i\} > 0$ be the minimum distance to the pacing-multiplier interval. Let $\|v\|_\infty := \max_i \|v_i\|_\infty$. It holds that*

$$\mathbf{E}(\bar{u}_i^t - u_i^*)^2 \leq \left( \frac{\|v_i\|_\infty^2}{\epsilon_i^2} + \left( \frac{1+\delta_0}{B_i} \right)^2 \right) \mathbf{E}(\beta_i^{t+1} - \beta_i^*)^2.$$

*Hence, letting $C = \frac{1}{(\min_i B_i)^2} \left( (\|v\|_\infty/\delta_0)^2 + (1+\delta_0)^2 \right)$, we have*

$$\mathbf{E}\|\bar{u}^t - u^*\|^2 \leq C \cdot \frac{(6 + \log(t+1))G^2}{(t+1)\sigma^2}.$$

Note that $C = \Omega(n^2)$. Hence, in this and the next theorems, the constant in the bound is $\Omega(n^4)$, which arises from $C$ and $\sigma = O(\min_i B_i) = O(1/n)$.

**Convergence of expenditures.** The *expenditure* of buyer $i$ at time step $t$ is $b_i^t = \beta_i^t v_i(\theta_t)\mathbb{I}\{i = i_\tau\}$. In other words, only the winner $i_t$ spends a nonzero amount, which is its bid. Let $\bar{b}_i^t = \frac{1}{t}\sum_{\tau=1}^t b_i^\tau$ be buyer $i$'s average expenditure. Utilizing the above convergence results, we show mean-squared convergence of $\bar{b}^t$ to $B$ at an $O((\log t)^2/t)$ rate.

**Theorem 5.** *For each $i$, it holds that*

$$\mathbf{E}(\bar{b}_i^t - B_i) \leq 2\left[ (\beta_i^*)^2 \mathbf{E}(\bar{g}_i^t - u_i^*)^2 + 2\|v_i\|_\infty^2 \frac{1}{t}\sum_{\tau=1}^t \mathbf{E}(\beta_i^\tau - \beta_i^*)^2 \right].$$

*For $t \geq 3$ and the constant $C$ defined in Theorem 4, we have*

$$\mathbf{E}\|\bar{b}^t - B\|^2 \leq \frac{2G^2}{t\sigma^2}\left( 6(C + \|v\|_\infty^2) + (C + 6\|v\|_\infty^2)\log t + \frac{\|v\|_\infty^2}{2}(\log t)^2 \right).$$

**PACE attains OME asymptotically.** Next, we show that PACE attains OME asymptotically, i.e., it generates allocations and prices that make buyers *no-regret* and *envy-free* in the limit (these notions will be clarified shortly). Let $x_i^t := \mathbb{I}\{i = i_t\}$ denote whether buyer $i_t$ is the winner (i.e., whether she is allocated the item $\theta_t$ at time $t$) Utilizing Theorems 4 and 5, we can show that buyer $i$'s regret, that is, the difference between the maximum possible utility in hindsight $\hat{U}_i^t$ (Definition 3) and the realized utility $\bar{u}_i^t$, vanishes as $t$ grows. The same holds for each buyer's envy. In other words, at a large $t$, in hindsight, no buyer prefers another buyer's set of allocated items (up to a vanishing error).[6]

**Theorem 6.** *Denote $\xi_i^t = |\bar{u}_i^t - u_i^*|$, $\Delta_i^t = |\bar{b}_i^t - B_i|$, $\gamma_t = \frac{\|v\|_\infty}{t}\sum_{\tau=1}^t \|\beta^\tau - \beta^*\|_\infty$. Let $r_i^t := \max\{\hat{U}_i^t - \bar{u}_i^t, 0\}$ be the* regret *of buyer $i$ at time $t$. Then, it holds that $r_i^t \leq \xi_i^t + \gamma_t/B_i$ and $\mathbf{E}(r_i^t)^2 = O\left((\log t)^2/t\right)$. Furthermore, define the* envy *of buyer $i$ (w.r.t. all other buyers) at time $t$ be $\rho_i^t = \max_k \bar{u}_{ik}^t/B_k - \bar{u}_i^t/B_i$, where $\bar{u}_{ik}^t = \frac{1}{t}\sum_{\tau=1}^t v_i(\theta_\tau)x_k^\tau$ is buyer $i$'s time-averaged utility given her own valuations and of buyer $k$'s allocations. Denote $\eta_i^t = \frac{1}{t}\sum_{\tau=1}^t (p^*(\theta_t) - \beta_i^\tau v_i(\theta_t))x_i^t$. It holds that*

$$\rho_i^t \leq \frac{1}{B_i}\left( \xi_i^t + \max_{k\neq i}\frac{\Delta_k^t + \eta_k^t}{B_k} \right) \quad \text{and} \quad \mathbf{E}(\eta_i^t)^2 \leq \frac{\|v\|_\infty^2 G^2}{t\sigma^2}\left( 6(1 + \log t) + \frac{(\log t)^2}{2} \right).$$

*Hence, the envy $\rho_i^t$ of buyer $i$ vanishes in mean square, i.e., $\mathbf{E}(\rho_i^t)^2 = O\left((\log t)^2/t\right)$.*

In light of Definition 3, Theorem 6 shows that $(x_i^\tau)_{(i,\tau)\in[n]\times[t]}$ is approximately optimal for buyer $i$. Since PACE also clears the market, we conclude that it attains OME asymptotically. Recall that theorem 4 ensures that buyers' $\bar{u}_i^t$ converge to their static equilibrium utilities $u_i^*$. Since the latter satisfy Pareto optimality and proportional share guarantee ($u_i^* \geq B_i$ for all $i$), so are the time-averaged realized utilities in the limit. Together with Theorem 6, we conclude that PACE achieves the said fairness and efficiency guarantees, namely, Pareto optimality, envy-freeness and proportional-share guarantee, asymptotically.

---

[6]In a static market, given an allocation $x \in \mathbb{R}_+^{n\times m}$, the (maximum, budget-weighted) *envy* of buyer $i$ toward others' bundles is $\rho_i(x) = \max_k \langle v_i, x_k\rangle/B_k - \langle v_i, x_i\rangle/B_i$ (see, e.g., [12, 46]). It is well-known that $\rho_i(x^*) = 0$ for all $i$ at equilibrium, a consequence of buyer optimality (Definition 2).

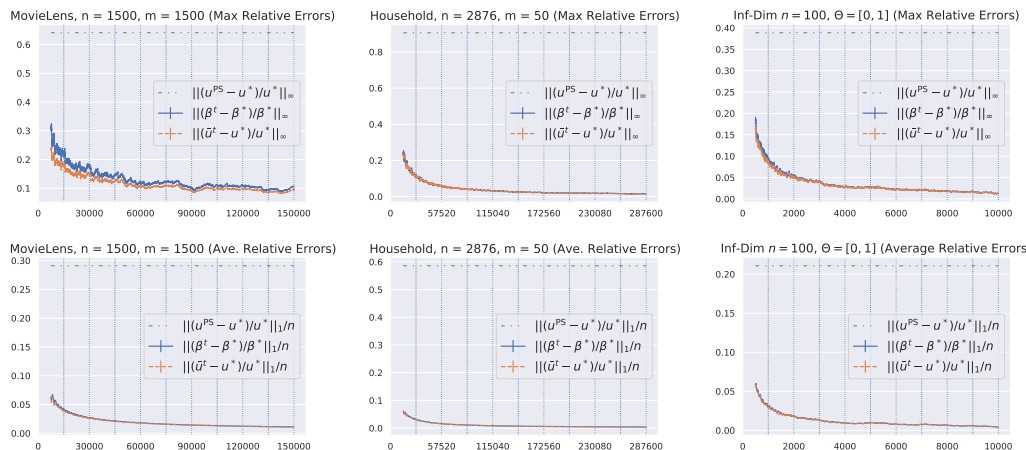

Figure 1: In all of our markets, iterates of the PACE dynamics quickly converges to their static equilibrium values both in the average case and the worst-off-buyer case. The horizontal line shows the fraction of $u^*$ achieved by the proportional share solution. The PACE utilities quickly outperform the proportional share utilities. Vertical lines indicate when $t$ is a multiple of $10n$.

## 6 Experiments

We evaluate the PACE dynamics in several real and synthetic datasets, namely, MovieLens, Household Items and an infinite-dimensional market instance with item space $\Theta = [0, 1]$ and $v_i$ being linear functions on $[0, 1]$. For the first two datasets, see [31] for more information and exploratory data analysis. For all datasets, we consider the CEEI (fair division) setting where $B_i = 1/n$ for all $i$. For each dataset (with number of buyers $n = 1500, 2876, 100$, respectively), we run PACE for $T = 10n$ time steps (iterations). More details on the experiments and additional plots displaying convergence of expenditures can be found in Appendix D. Figure 1 displays the mean values of the average and maximum relative errors of the pacing multipliers and time-averaged cumulative utilities over 10 repeated experiments with different seeds (relative errors of cumulative spending w.r.t. total budgets are plotted separately in Appendix D). The standard errors are also displayed as vertical bars but are very small and nearly invisible. Vertical dotted lines indicate $t = 10n$ The figures do not show the initial iterates $t = 1, \ldots, 5n$.

We see that PACE converges very quickly numerically: within 10 epochs ($10n$ time steps) average deviations in most quantities falls within 5% of the equilibrium quantity, with the worst case not far behind. An important point is that budget spend takes much longer to converge than utility. This demonstrates an important practical difference for using PACE in an allocation scenario where budgets are 'real money' (e.g. Internet ad impressions) as compared to a CEEI-like setting, where budgets are faux currency only used for fair division.

## 7 Conclusion

We introduced the concept of an online Fisher market and proposed the PACE dynamics. We showed that when items arrive sequentially and stochastically, PACE converges to equilibrium outcomes of the underlying market model. Furthermore, we showed that, as a consequence of this, PACE can be used in online fair division problems to generate an online allocation that, asymptotically, achieves the compelling fairness properties of CEEI.

Many questions remain for future research. We mostly focused on the case where budgets are faux currency and there are many open questions for adapting PACE to a real-money budget-management setting as well as more complicated nonlinear utility models. Another imperative question, especially for practitioners, is whether PACE guarantees some level of incentive-compatibility.

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

**Checklist.** We answered the questions as accurately as possible and believe that many of them do not apply to our work. We are happy to elaborate further should there be any questions or concerns.

