# OpenReview forum: "Online Market Equilibrium with Application to Fair Division"
_NeurIPS.cc/2021/Conference — NeurIPS 2021 Poster_

### Official Review · Reviewer_pkgu · 2021-06-28

**Rating:** 5
**Confidence:** 3

**Summary:**

The submission considers a setting of static buyers and online arriving items from an i.i.d. distribution. Buyers have linear utilities for items and budget constraints that only have to be satisfied in expectation.

The authors propose running a centralized first-price auction, together with a simple distributed algorithm for adaptively adjusting a pacing multiplier for the buyers’ bids. They show that after a sufficiently large number of rounds (polynomial in the number of buyers), their procedure converges to the static equilibrium outcome with all its nice properties.


**Main Review:**

Understanding distributed convergence to approximate equilibria in online settings is an important problem. I’m not very clear on how novel the current submission is compared to related work, e.g. on pacing equilibria.

[EDIT: The above concern has been mostly addressed by authors in rebuttal.]

I’m also pretty annoyed with the way the dependence on number of buyers is hidden. The introduction promises convergence rate O(log(t)/t), and claims that the method is appropriate for “internet-scale”. Then Theorem 3 has a quadratic dependence on \sigma, which is basically \kappa, which is basically the $n$. Theorems 4 and 5 has an additional factor called C, which is again \sigma^2 (so n^4 in total if I’m not mistaken). The guarantee of Theorem 6 has parameters \zeta and \delta which are “as in Theorem 6”…

I think that the paper would benefit from being written in a simpler setting with discrete items and much less notation.



**Time Spent Reviewing:**

1

---

> ### Author Response · Authors · 2021-08-10
> **Response to Official Review of Paper6333 by Reviewer pkgu**
>
> Thanks for the questions and comments.
>
> **Novelty w.r.t. pacing equilibrium**
>
> Pacing equilibrium, as discussed in references [18,19] of our paper, is an \emph{offline} solution concept, and thus incomparable with what we consider here. Our PACE algorithm, applied to an **online** quasilinear market, converges to the online analogue of a first-price pacing equilibrium (FPPE) as defined in [19]; see lines 239-241 and Appendix C.
> As the reviewer agrees, understanding distributed convergence to approximate market equilibria in an online setting is an important problem. To the best of our knowledge, our algorithm is the \emph{first} online algorithm for computing FPPE in an online fashion. This is also pointed out in lines 17-19 & 81-82 in our paper. Furthermore, it is also simple, highly distributed across buyers, and interpretable as an auction-like mechanism.
>
> **...a simpler setting with discrete items…**
>
> We started writing the paper using a discrete item space but realized that discrete & continuous lead to similar complexity of results. Here, using a continuous/measurable item space is indeed meant to address Internet-scale problems, which usually consist of a huge number of items and a moderately large number of buyers. In this case, modeling it as a continuous space allows parametrization via a small number of features. We will make sure to elaborate on this when describing the setting.
>
> More importantly, each iteration of the algorithm costs O(n) time and is fully distributed among the n buyers; it is extremely simple and decentralized; no stepsize needed. This does \emph{not} depend on item space complexity at all.
>
> **Dependence on #(buyers)**
>
> As mentioned above, each iteration costs O(n) and is distributed among the buyers, apart from the step to find the winner using paced bids; it does not depend on item space complexity. This makes PACE highly scalable in both n and #(items), where there is no explicit dependence on the latter. Numerically (Sec 6), PACE iterates quickly converge to their equilibrium counterparts. We will make sure to point out the dependence on $n$ in the bounds in the texts. Also note that the constants in bounds in Theorems 3-6, as in many convergence results of first-order methods, often do not fully capture their practical performance.
>
> We hope the above clarifications clear some doubts about the work and hope the reviewer can reconsider the evaluation.

---

> > ### Comment · Reviewer_pkgu · 2021-08-13
> > **Pacing equilibria and novelty**
> >
> > Thanks for the clarification! So if I understand correctly, previous work on static pacing equilibria established that it can be solved with EG program, and you show how to do this online. The online case sounds very natural - I'm happy to raise my score.
> >
> > Reading the other reviews, I would like to understand better how your work relates technically to Xiao [48] and Gao and Kroer [24].

---

> > > ### Author Response · Authors · 2021-08-15
> > > **Re: Pacing equilibria and novelty**
> > >
> > > Thanks for the reply and reconsideration.
> > >
> > > The starting point of our work is to apply dual averaging (DA) [48] to solve the dual EG convex program in [24] (but note that this convex program is an offline problem, unlike our setting). We made a slight extension of the technical setting of [24], requiring only a measurable item space instead of a subset of a Euclidean space. But there is a lot more beyond that.
> > >
> > > First, we propose the concepts of an online Fisher market and an equilibrium, a problem setting independent of any convex program or any stochastic optimization algorithm. In this setting, we show that a market equilibrium can be achieved asymptotically by running PACE, as long as the items are drawn i.i.d. from a (hidden, unknown) distribution. We also give an example that shows no algorithm gives any meaningful guarantee if item arrivals are adversarial (line 305, footnote 5 and Appendix, lines 604-648).
> > >
> > > Second, we further interpret/rewrite the application of DA to dual EG as an auction-like mechanism (i.e., PACE) and show that the updates can be done quickly and in a distributed manner. Lines 205-214 first introduce PACE and lines 285-302 show how it is derived from DA. In particular, the subgradient and proximal steps (lines 261 & 263) translate to simple and interpretable operations of finding the winner and buyers updating their pacing multipliers, respectively. These are the key observations that make the algorithm practical.
> > >
> > > Furthermore, the convergence results for utilities, expenditures, regret and envy regarding PACE (Theorems 4-6) are entirely new. In particular, they do not follow directly from any results in [24] and [48] and are established largely from scratch.

---

> > ### Comment · Reviewer_pkgu · 2021-08-13
> > **Rate of convergence and complexity**
> >
> > Although your algorithm looks plausibly applicable in practice, I'm still disturbed by the rate of convergence (and even more disturbed by how that's **hidden in the current writeup!**)
> >
> > In particular, *in theory* you could take an algorithm with very slow iterations, break the iterations into sub-iterations and hide the complexity in the rate of convergence. Having more iterations is actually *worse* than slower iterations because it requires more interaction/adaptivity.

---

> > > ### Author Response · Authors · 2021-08-15
> > > **Re: Rate of convergence and complexity**
> > >
> > > The bounds in Theorems 3-5 are completely explicit in both $n$ and $t$, since the constants are defined right there. It is not being hidden in big-O notation or any such obfuscation. Nonetheless, we see why the reviewer feels that the bounds could be clearer in terms of the dependence on $n$; particularly due to normalizing the budgets to sum to one, which was done for convenience.
> > >
> > > As already stated in our rebuttal above, we are happy to emphasize further the dependence on $n$. When we refer to the convergence rates in big-O notation in $t$ in the texts (e.g., line 77), we will make sure to state the dependence on $n$ as well.
> > >
> > > Re “Having more iterations by splitting up updates”. We suppose this is something that one could do, but it really does not have anything to do with our paper (and frankly it’s a bit of a strawman): the PACE algorithm (or any algorithm for similar purposes; unless you do stochastic/block-coordinate updates on the buyers) needs to involve all $n$ buyers in an iteration in order to update parameters. This necessarily incurs $O(n)$ cost per iteration, which is the cost per iteration we achieve. Here, the constants in $O(n)$ are small, absolute constants arising from basic arithmetic operations: see lines 203-214 describing PACE. The reason why this is attractive is that something like a second-order method would require ~$O(n^{3.5})$ cost *per iteration*, which means that progress is made very slowly. Secondly, we would like to emphasize that there was previously no algorithm whatsoever for the online setting we consider, and the numerical performance of PACE is quite strong.

---

### Official Review · Reviewer_3dLo · 2021-07-09

**Rating:** 7
**Confidence:** 4

**Summary:**

This paper investigates the problem of finding market equilibrium in an online Fisher market. Different from the traditional static Fisher market, in the online setting the items arrive one by one following an unknown distribution. The algorithm needs to allocate the items in the same online fashion with the goal that the time-averaged allocation and utilities asymptotically converge to an online market equilibrium. The main result of this paper is a simple PACE algorithm that achieves this goal. The algorithm maintains a pacing variable for each buyer and at each step assigns the item to the buyer with the highest paced value, after which the pacing variables are updated accordingly. In the essence, this PACE algorithm is a version of the dual averaging algorithm applied to the dual of the EG convex program, and therefore enjoys many nice converging properties. The authors also discuss the implication of this PACE algorithm to the online fair division problem and extend it to the Fisher market with a continuum of items. Finally, experiments are conducted to show its quick convergence in practice.

**Limitations And Societal Impact:**

Yes.

**Main Review:**

This paper studies a very natural and well-motivated problem of online resource allocation and proposes a simple dynamic with nice converging properties. The results are strong, in the sense that the dynamic is simple and intuitive, and does not need to know the underlying item arriving distribution and can converge to a market equilibrium with fast converging time. The paper is also well-written, with clear intuition and explanation provided at each step.

The idea of the PACE dynamic is rather straightforward once its connection to the dual averaging dynamic is established. As a result, the converging proof is not particularly innovative. Although I understand it is also not trivial and a lot of details need to be taken care of to ensure the convergence of the time-averaged utilities, expenditure, allocation and realized prices to the equilibrium at the same time.


**Time Spent Reviewing:**

6

---

### Official Review · Reviewer_P1UM · 2021-07-16

**Rating:** 7
**Confidence:** 4

**Summary:**

=Paper summary=
The authors study an online Fisher market. Unlike classic Fisher markets, here the items arrive one by one in an online fashion and must be immediately and irrevocably allocated to a set of static agents. The online market equilibrium notion introduced is a pair of sequences of (step-wise) allocations and prices, such that the market clears and the agents have no regret in hindsight. The authors present an algorithm for allocating the items, named PACE dynamics, with the property that the prices and the utilities converge to those of an equilibrium as the number of time steps goes to infinity. PACE is essentially the dual averaging algorithm on a regularized convex program that is the reformulation of the dual of the generalized Eisenberg-Gale convex program of Gao and Kroer [24]. Experiments on real datasets are also presented, showcasing that PACE converges fast in practice.


**Limitations And Societal Impact:**

Yes

**Main Review:**

=Strengths - Weaknesses=

The paper is well-written, although it is a bit dense and notation-heavy at times. The topic is an interesting twist in a classic economics model that is well-motivated by modern applications. There is novelty in the model itself and the proposed dynamics is not just a direct application of dual averaging. Given other recent results in asymptotic online fair division (see below), a guarantee similar to that of Theorem 6 (and thus the ones of Theorems 3-5) is probably the best one could hope for. The paper is clearly technical and the authors did try to make the submission self contained in that respect. That being said, it is also a fact that this work relies a lot on the heavy machinery of Xiao [48] and Gao and Kroer [24]. Also, as all the proofs are in the supplementary material, I only read the proofs of Theorems 3, 4 and 5.
While I have some concerns about NeurIPS being the right venue for this work (mostly in terms of presentation opf the content), this is a solid paper.


=Comments for authors=

-- There is recent work on vanishing envy in online fair division that should be included and discussed. See, e.g., the following papers and the references therein:
* Gerdus Benade, Aleksandr M. Kazachkov, Ariel D. Procaccia, Christos-Alexandros Psomas: How to Make Envy Vanish Over Time. EC 2018: 593-610
* David Zeng, Alexandros Psomas: Fairness-Efficiency Tradeoffs in Dynamic Fair Division. EC 2020: 911-912

-- line 154, “The item must be allocated irrevocably to *one* buyer.”: While this is what eventually happens in PACE, my understanding is that this is not part of the description of the model. Please clarify.

-- The $B_i$s were a constant source of confusion for me. In fact, things only got clear after reading the last supplementary section of the appendix. In particular, it is never stated explicitly in the main paper what the $B_i$ are. It is not even clear whether they are the total or the per-step budget of agent $i$, until fully understanding Definition 3. The informal discussion before Definition 3 talks about “per-period expenditure rate” but a “period” is never defined either. Also, I assumed that being the budget per step, there would be some kind of per step budget constraint and that the remaining budget of a step $t$ would be transferred to the budget of step $t+1$. Apparently this is not what happens, as $B_i$ is interpreted as a (very) soft constraint. Both this and a lower bound on $B_i$ should be discussed.

--In connection to a discussion about a lower bound on $B_i$, if I am not mistaken, currently there is no restriction on $B_i$ being a function of $t$, say $O(1/t)$, something that would completely break down the convergence bounds. So maybe it should be stated explicitly that $B_i$ is independent of the step and is something like $\Omega(1/n)$.

-- Is there a sufficiently small value for $\delta_0$? I did not find the comment in appendix B very clear. It seems like any $\delta_0 > 0$ should work.

-- line 262: the last term should be $\frac{1}{t} g^t$ (i.e., the overbar should be missing).

-- Maybe it would be useful to define the strong convexity modulus of a convex function.

-- line 342: “where … as in Theorem 6” can be removed as this is Theorem 6.

-- line 355: I believe this should be $T=100n$.


=Questions to authors=

If possible, I would like to see a response to my 2nd, 4th, and 5th comments above.


------------ AFTER REBUTTAL ------------
I have read the other reviewers' comments and the authors' rebuttal. I appreciate the effort the authors put into this. My comments were addressed adequately and I remain positive about this submission.

**Time Spent Reviewing:**

6.5 hours

---

> ### Author Response · Authors · 2021-08-10
> **Response to Official Review of Paper6333 by P1UM**
>
> Thank you for the detailed comments. We will take all of them into consideration and make sure to correct the typos. We will certainly discuss the suggested recent works.
>
> [2nd]
> You are right, we don’t need it in the model description and fractional allocations can be allowed. It is just natural to think of arriving items as indivisible ones, which is also closer to reality (such as ad slots). While PACE as described in the paper will still assign each item to a single buyer, allowing fractional allocations / randomization in tie-breaking does not affect the guarantees at all. From the dual averaging perspective, these are in fact just different subgradient oracles. We will mention this.
>
> [3rd & 4th]
> We view the budgets $B_i$ as inputs to the problem, along with the valuations $v_i$; hence, we do not consider the case of changing $B_i$ over time. The budgets can be any positive quantity. We will make sure to state it explicitly. We assume the sum of all $B_i$ is 1 to simplify constants in the analysis; it is not necessary for the algorithm to run. As in offline Fisher market equilibria, scaling up all buyers’ budgets simultaneously by the same constant changes prices and utility prices ($\beta$) accordingly, all at the same time; allocations are not affected.
>
> [5th]
> Yes, any $\delta_0>0$ works in theory (it affects the constants in the bounds, e.g., in Theorem 4). In experiments, we observe that a smaller value (between 0 and 1) sometimes leads to slightly faster numerical convergence. Overall, we conclude that its value does not have a significant effect on numerical convergence. We used $\delta_0=0.05$ in the experiments.

---

### Official Review · Reviewer_ZYd4 · 2021-07-17

**Rating:** 6
**Confidence:** 4

**Summary:**

The paper studies bidding in the first pice auction with budget constraints. In this problem, there are n bidders/buyers and T rounds of auctions. In each round t, an item arrives and the bidders observe their value for the item and need to submit a bid for it. The paper studies if one can design a bidding policy that results in an online market equilibrium (OME). The authors present a pacing algorithm in which the submitted bid of a buyer is equal to the buyer's pacing factor and his/her bid. The pacing factor is then get updated over time. It is shown that under the pacing algorithm, the pacing factors, which are the dual variable of the budget constraints, converge. In addition,  the buyers' time-averaged utility converges to the optimal equilibrium utility and the buyers' time-averaged expenditure converges to the target budget range.

**Ethics Review Area:**

["I don’t know"]

**Limitations And Societal Impact:**

see above

**Main Review:**

The idea of pacing is based on the literature on online convex optimization and in particular online descent algorithms. This idea has been used in reference [7] of the paper for second-price auctions.
 ([7] S. R. Balseiro and Y. Gur. Learning in repeated auctions with budgets: Regret minimization
and equilibrium. Management Science, 65(9):3952–3968, 2019.)
The authors did not cite and discuss [7] properly. Note that [7] has nice equilibrium results for stochastic and adversarial markets. At the high level [7] also designs a pacing algorithm in which the pacing factors (dual variables of budgets constraints) get updated over time. Hence, it is not clear if the current paper has a significant contribution.

Another point is about presenting the results. At the end of the day, the authors solve bidding in the first pice auction with budget constraints. But, for some reason, the authors avoid discussing their problem in this form and this can cause confusion. In particular, the intro is not very to the point.

Finally, the paper may not be the best fit for this conference. EC  for example might be a better fit.

**Time Spent Reviewing:**

3 hours

---

> ### Author Response · Authors · 2021-08-10
> **Response to Official Review of Paper6333 by Reviewer ZYd4**
>
> Thanks for the questions and suggestions.
>
> **Regarding [Balseiro & Gur]**
>
> While we are happy to add more discussion on B&G, we would like to clarify that the work of B&G and our paper are very different in terms of problem settings, results, and analysis.
>
> In particular
>
> - B&G is in a second price auction setting with real budgets, we are in a fair division setting where budgets are “faux currency” simply created by the mechanism. The fact that the solution to the stochastic fair division problem in our paper can be framed as repeated first price auctions is a result, not an assumption.
> - Relatedly, B&G study approximate Nash equilibrium properties while we are interested in fair division properties (e.g. no envy, proportional share guarantees, maximization of Nash welfare).
> - To achieve convergence to approximate Nash equilibrium, B&G require a restrictive monotonicity assumption that is hard to verify (Assumption 4.1 in their paper), we require nothing like that.
> - B&G shows ergodic convergence, a much weaker notion than our proof of last iterate convergence
> - B&G show ergodic convergence of regrets at a $1/\sqrt{T}$ rate, and eventual ergodic convergence of the pacing multipliers (with no guarantee on the rate). In contrast, our result (Theorem 4) gives **last-iterate** convergence of both pacing multipliers and regrets at a faster $1/T$ rate.
> - The B&G algorithm requires step-sizing, we do not.
>
> [More on B&G stepsize rules v.s. no stepsize in our setting] The analysis of B&G builds upon mirror descent and a stochastic approximation framework. Hence, it requires pre-determined vanishing stepsizes (their Assumption 1), even in the simultaneous learning setting (their Assumption 3); stepsizes of all bidders also need to be carefully selected for implementation. In contrast, our algorithm PACE is completely stepsize free, thanks to the dual EG convex program, the dual averaging framework, and our leveraging strong convexity of the dual objective. This makes it vastly easier to apply our algorithm in practice. Finding the right \emph{joint} stepsizes for all bidders in a large-scale system may be difficult, error prone, and may lead to excessively conservative stepsizes. Our algorithm has no such issue: it can be run as is.
>
> Hopefully the above makes it clear that the two papers are quite different. While the results of B&G are very nice for their setting, they are much weaker than what we achieve for our problem. The main similarity is that both use the well-known idea of pacing. We will make sure to point this out.
>
> **Regarding scope**
>
> While EC is indeed another good venue, we believe this paper is also a good fit for NeurIPS - our work fits squarely into “Theory - Algorithmic Game Theory” of the official areas of research listed here: https://neurips.cc/Conferences/2021/CallForPapers as well as the “Optimization - convex optimization” area.
>
> **Regarding presenting the results**
>
> Although our PACE algorithm can be viewed as first-price auction dynamics, our main focus is on online Fisher markets, motivated by problems such as online fair allocation, where buyer budgets are in terms of funny money, not real money. While there are some deep connections to auctions, market equilibrium concepts and their convex optimization characterizations belong to another research area and are of independent interest. In particular, market equilibrium properties such as no-envy, Pareto optimality, and proportionality are completely different from game-theoretic properties such as approximate Nash equilibrium.
>
>
> Given the above clarifications, we sincerely hope the reviewer can reconsider the evaluation.

---

### Decision · Program_Chairs · 2021-09-27

**Decision:**

Accept (Poster)

**Comment:**

I decided to accept the paper since it presents an interesting problem and provides a novel solution with a surprising connection to the dual averaging technique.
In the final version, please make sure to clearly and explicitly discuss the bounds and their dependence on the problem parameters.